# Spatial variation and predictors of missing birth preparedness and complication readiness (BPCR) messages in Ethiopia

**Aklilu Habte**[1]*, **Samuel Hailegebreal**[2], **Tamirat Melis**[3], **Dereje Haile**[4]

**1** School of Public Health, College of Medicine and Health Sciences, Wachemo University, Hosanna, Ethiopia, **2** Department of Health Informatics, College of Medicine and Health Sciences, School of Public Health, Wachemo University, Hosaena, Ethiopia, **3** Department of Public Health, College of Medicine and Health Sciences, Wolkite University, Wolkite, Ethiopia, **4** Department of Reproductive Health, College of Medicine and Health Sciences, School of Public Health, Wolaita Sodo University, Sodo, Ethiopia

* akliluhabte57@gmail.com

## Abstract

**Data Availability Statement:** The data for this study were obtained from the DHS program with a reasonable request. Thus, the one who needs the data supporting the findings of this study can get it

### Background

The Birth Preparedness and Complication Readiness (BPCR) message is one of the prenatal care packages targeted at reducing maternal and neonatal mortality by avoiding unnecessary delays during labor and delivery. There is limited evidence in Ethiopia that has looked at the spatial variation of missing BPCR messages and potential predictors. Hence, this study aimed to identify spatial predictors missing BPCR messages at the national level.

### Methods

The study was based on analysis of 2016 Ethiopia Demographic Health Survey data, using a weighted sample of 4771 women. Arc-GIS version 10.7 and SaTScan version 9.6 statistical software were used for the spatial analysis. To explore spatial variation and locate spatial clusters of missing BPCR messages, the Global Moran's I statistic and Bernoulli-based spatial scan (SaTScan) analysis were carried out, respectively. Hotspot (Getis-OrdGi*) analysis was conducted to identify Hotspots and Cold spotsof missing BPCR messages. Finally, spatial regression were carried out via ordinary least squares and geographically weighted regression to identify predictors of hotspots for missing BPCR messages.

### Results

The overall prevalence of missing BPCR messages in Ethiopia was found to be 44.0% (95% CI: 42.6, 45.4%), with significant spatial variation across regions (Moran's I = 0.218, p-value<0.001) and seven most likely significant SaTScan clusters. The vast majority of Somali, central Afar, and Gambella regions were identified as statistically significant hotspots. Living in the poorest wealth quintile, having only one ANC visit, lack of access to listening to the radio, facing difficulty in accessing money, not having a mobile phone, and being not covered by health insurance were identified as significant spatial predictors of missing BPCR messages.

in anonymized form from the DHS website at https://www.dhsprogram.com upon reasonable request in the same manner as the authors did.

**Funding:** The author(s) received no specific funding for this work.

**Competing interests:** The authors have declared that no competing interests exist.

**Abbreviations:** AIC, Akaike's information criterion; ANC, Antenatal Care; BPCR, Birth preparedness and complication readiness; CSA, Central Statistical Agency; EDHS, Ethiopian Demographic and Health Survey; GWR, Geographically Weighted Regression; OLS, Ordinary Least Square; WHO, World Health Organization.

## Conclusion

The level of missing BPCR messages during pregnancy was found to be high in Ethiopia, with significant local variation. As a result, policymakers at the national level and local planners should develop strategies and initiatives that enhance women's economic capacities, health-seeking behavior, and media exposure. Furthermore, the regional authorities should focus on strategies that promote universal health coverage through enrolling citizens in health insurance schemes.

## Introduction

Despite a decline in the global maternal mortality ratio (MMR) from 339 in 2000 to 223 in 2020, it remains unacceptably high, with nearly 287,000 women dying during and after pregnancy and childbirth [1, 2]. Almost 95% of all maternal deaths occurred in Low and Middle-Income Countries (LMICs) in 2020, and most could have been prevented [3]. Sub-Saharan Africa (SSA) and Southern Asia accounted for approximately 70 percent (202,000) of the estimated global maternal deaths in 2020 [1, 2]. Between 2000 and 2016, Ethiopia reduced maternal death from 871 to 412 per 100,000 live births and child mortality from 166 to 67 per 1,000 live births, still, both remain much too high [4, 5].

Delays in preparing for and responding to potential direct obstetric complications during pregnancy and childbirth by pregnant women, families, and healthcare providers have been recognized as the key barrier to reducing maternal morbidity and mortality in SSA [6]. Thus, interventions such as the Birth preparedness and complication readiness (BPCR) package are vital for addressing such delays in developing countries like Ethiopia, where obstetric services are often limited and/or underutilized [7].

BPCR is one of the WHO recommendations for newborn and maternal health promotion activities that are delivered and executed through an antenatal care (ANC) service [8]. It is defined as a comprehensive strategy to increase the adoption and effectiveness of essential maternal and newborn health care by reducing three delays in accessing basic maternal health services (delays in deciding, reaching, and receiving care) [9]. It has a set of indicators that were intended to be implemented by pregnant women, families(husband/partner), communities, health facilities, and health care providers [10]. Those items of care that are intended to be implemented by all pregnant women are: deciding on a birthplace, maintaining the necessary supplies for childbirth, figuring out or establishing emergency transportation, setting aside money or an emergency fund, identifying blood donor/s, determining companions during labor and childbirth which are vital in easing access to care [10, 11].

Despite having a considerable impact on maternal mortality reduction, BPCR messages were not widely implemented in Ethiopia. A systematic review and meta-analysis conducted in Ethiopia revealed that BPCR uptake was 25.2% [11]. On the other hand, a recent study from nationally representative data showed that more than half (56.02%) of women received at least one BPCR message [12]. Small-scale studies conducted in the country showed that there was a positive association between the receipt of BPCR messages and educational status, occupation, being in a higher wealth quintiles, receipt of ANC visits, previous history of health facility delivery, history of adverse birth outcomes, number of living children (parity), knowledge of key danger signs during pregnancy and postpartum, and male partner involvement in BPCR plan [13–18].

Despite numerous studies on the uptake of BPCR messages across the country, there was a dearth of evidence that has investigated the spatial distribution that shows regions with poor

uptake of BPCR messages. Hence, the current study carried out both spatial analytic techniques (a hot spot and a geographically weighted regression (GWR)). First, a hotspot analysis was carried out to identify the regions and areas with low BPCR message adoption, and the study will next delve into identifying potential variables that lead to regional differences in poor service uptake by employing a GWR. Conducting a hotspot analysis could offer a data-driven approach to understanding and addressing the challenges hindering the dissemination and adoption of key BPCR messages. By identifying specific geographic regions with low BPCR message adoption, resources and interventions can be allocated to areas of highest need and groups that are especially vulnerable to maternal complications of pregnancy. The findings from the current study will allow program planners, policymakers, and healthcare providers to customize their efforts to address the factors behind poor service uptake in each hotspot, thereby increasing the likelihood of successful intervention. By targeting interventions in hotspots with low BPCR message uptake, maternal and neonatal health outcomes can be significantly improved via working on safer pregnancies, and timely access to healthcare services.

## Methods and materials

### Study design and setting

The current study was based on the 2016 Ethiopian Demographic and Health Survey (EDHS) which was conducted by the Central Statistics Agency (CSA) from January 18, 2016, to June 27, 2016 [5]. Ethiopia is located in the Horn of Africa (3˚–15˚N latitude and 33˚–48˚E longitude). The country is divided into nine administrative regions (Tigray, Afar, Amhara, Oromia, Somali, Benishangul-Gumuz, Southern Nation Nationality and People's Region (SNNPR), Gambella, and Harari) and two self-administrative cities (Addis Ababa and Dire Dawa) [5]. A secondary analysis of the women's (IR) file of 2016 EDHS was carried out.

### Populations

The source populations were all women aged 15 to 49 who gave birth within the five years preceding the survey, whereas the study populations were women who had complete information on the uptake of BPCR messages during their ANC visit. The EDHS has numerous datasets, for males, women, children, birth, and households. The women's (IR file) was used for the current study, and a total weighted sample of 4,771 women was considered in the analysis.

### Sampling procedures and data collection tool

The Ethiopia Population and Housing Census (PHC), carried out in 2007 by the Ethiopia CSA, served as the sampling frame for the 2016 EDHS. The samples were chosen using a stratified, two-stage cluster sampling procedure in which data were hierarchical. In the first stage, 645 enumeration areas (EAs) were selected (202 in urban and 443 in rural areas), with a probability proportional to EA size and independent selection in each sampling stratum. In each of the selected EAs, a household listing operation was carried out, and the consequent lists of households served as a sampling frame for the selection of households in the second stage. In the second stage, a fixed number of 28 houses per cluster were chosen with equal probability from the sample frame [5].

The Woman's Questionnaire, which includes socio-demographic and economic information, obstetric characteristics, and maternal health service utilization, was used to collect data from all eligible women aged 15 to 49. The location data (geographic coordinates) of each survey cluster were collected using Global Positioning System (GPS) receivers. Each cluster's GPS reading was taken in the center. To protect respondents' privacy, GPS latitude/longitude

positions for all survey groups were randomly displaced. The highest displacement for urban clusters was two kilometers (km), whereas, for 99% of rural clusters, it was five kilometers (km). The remaining 1% of rural clusters have been relocated up to a distance of ten km [19].

## Measurement of variables of the study

The outcome variable was missing BPCR messages and was measured based on the six key WHO-recommended messages delivered to a mother during her last pregnancy. The key messages were about deciding the place of birth, collecting the necessary supplies for childbirth, arranging emergency transportation, setting money aside for emergency conditions, identifying companions during labor and childbirth, and securing potential blood donors. Information on these six key BPCR messages was obtained from the response to the question: During any of your antenatl visist, were you told about your Place of birth? Were you told about supplies needed for birth? Were you told about emergency transportation? The responses were recorded as Yes (= 1) or No (= 0). A single woman may be advised about any of the six messages many times throughout the same pregnancy, but each response was recorded as a single message. Based on the responses, a composite index of BPCR was created, which had a minimum and maximum value of 0 and 6 respectively. The women with '0' values were regarded as 'missing BPCR messages,' whereas those with one or more messages were considered to have received BPCR [12, 20].

**Explanatory variables.** Potential determinants of BPCR were extracted from the data set after reviewing related and current literature [10–12, 18, 21] (Table 1).

**Autonomy in decision-making** was measured using replies to questions about who makes the ultimate decision for the family on big property purchases, visits to relatives, and health care. (i) respondent alone, (ii) respondent and husband/partner, (iii) husband/partner alone, (iv) someone else, and (v) others were the response categories. For each question, replies (i) or (ii) earned a score of 1, indicating good decision-making capacity, while the remaining responses were assigned a value of 0, indicating limited capacity. The responses on each of the three aspects were added together to get a total score ranging from 0 to 3. Finally, a composite score had been divided into two unique groups: low and high for "0 to 2" and "3" scores, respectively [22, 23].

## Data management and statistical analysis

STATA version 14.1 software was used for data extraction, cleaning, coding, and analysis, while ArcGIS 10.7 and SaTScan were used for spatial analysis. The data were weighted by applying a weighting factor $\left(\frac{v005}{1000000}\right)$ to minimize under- or over-representation of the data in the surveys due to differential selection among strata. Using the *svyset* command, the data was further structured as survey data. The weighting procedure was thoroughly discussed in the 2016 EDHS report [5]. The weighted proportions of a response and explanatory variables have been estimated in STATA and exported to Microsoft Excel 2016 before being loaded into ArcGIS 10.7 for further analysis. To describe the background characteristics of respondents, descriptive statistics such as frequency and percentage were used.

## Spatial analysis

**Spatial autocorrelation.**   To figure out if the spatial distribution of not receiving BPCR messages in Ethiopia was dispersed, clustered, or randomly distributed, the global spatial autocorrelation (Global Moran's I) was carried out [24]. The Global Moran's I statistic is used to quantify spatial autocorrelation by taking the whole data set and producing a single output

**Table 1. Potential determinants extracted from the EDHS 2016 and supposed to influence the uptake of BPCR messages.**

| Variables | Description | Category |
|---|---|---|
| Age | The respondent's age, expressed in years, at the time of the survey. | 1. 15–19, |
| | | 2. 20–34 |
| | | 3. 35–49 |
| Marital status | Percentage of women according to the current status of marriage or cohabitation. | 1. Married |
| | | 2. Never married |
| | | 3. Others |
| Women educational attainment | Percent distribution of women ages 15–49 by the highest level of schooling attended or completed. | 1. No education |
| | | 2. Primary |
| | | 3. Secondary |
| | | 4. Higher |
| Residence | The area where respondents lived when the survey was conducted. | 1. Urban |
| | | 2. Rural |
| Family size | Number of household members at the time of data collection | 1. $\leq$5 |
| | | 2. >5 |
| Sex of head of Household | Percent distribution of households by sex of head of household | 1. Female |
| | | 2. Male |
| Wealth index | Calculated using easy-to-collect data on a household's ownership of selected assets, such as televisions and bicycles; materials used for housing construction; and types of water access and sanitation facilities. | 1. Richest |
| | | 2. Richer |
| | | 3. Middle |
| | | 4. Poorer |
| | | 5. Poorest |
| Parity | The number of living children the woman had at the time of the survey | 1. Nulliparous |
| | | 2. Primiparous |
| | | 3. Multiparous |
| | | 4. Grand multiparous |
| Pregnancy status during last childbirth | Percentage of births to women aged 15–49 in the five years preceding the survey, including current pregnancies, by birth planning status—(i) wanted then, (ii) wanted later, or (iii) not wanted at all. | 1. Unwanted (*ii& iii)* |
| | | 2. Wanted (*i*) |
| Media exposure | The number of women aged 15 to 49 who are exposed to specific media at various frequencies, such as reading a newspaper, watching television, and listening to the radio. | 1. Not at all |
| | | 2. Less than once a week |
| | | 3. At least once a week |
| Autonomy in decision-making [a] | The total number of currently married women aged 15 to 49 who make decisions about their own health care, big household purchases, and visits to family or relatives. | 1. Low |
| | | 2. Medium |
| | | 3. high |

value ranging from -1 to +1. Moran's I value at -1, +1, and 0 indicates the likelihood of missing BPCR messages being dispersed, clustered, and randomly distributed, respectively. The Z-score determines the statistically significant difference in clustering, and the p-value determines the significance. Accordingly, Moran's, I value was statistically significant ($p<0.05$), showing the presence of spatial autocorrelation and it showed the need for hot spot analysis [25].

**Incremental spatial autocorrelation.** The peak distance from the spatial incremental autocorrelation model identifies the place where the spatial dependency of missing BPCR

messages was most evident and that highest peak distance value has been selected as the threshold distance for hotspot analysis (**S1 File**) [26].

**Hot spot analysis (Gettis-Ord Gi\* statistics).**   The local spatial analysis was performed using Getis-Ord Gi\* statistics to identify important hot spot and cold spot areas for missing BPCR messages. The hotspot analysis compares the local mean rate (the rates for a cluster and its nearest neighbouring clusters) to the global mean rate (the rates for all clusters). The z-score was determined to confirm the statistical significance of clustering, and the p-value was calculated to establish the significance at a p-value<0.05 with 95% CI. Cold spots were announced if the z-score is less than -1.96, and hotspot locations were declared if the z-score was greater than +1.96 [24, 27–29]. In addition, Incremental Spatial Autocorrelation was examined to determine a suitable threshold for finding spatial processes that favour clustering [30].

**Spatial interpolation.**   Based on reported or observed values from selected enumeration areas (EAs), spatial interpolation was used to anticipate the proportion of missing BPCR messages in unsampled EAs. For doing this, the standard Kriging interpolation method was used instead of other interpolation approaches since it is an optimal interpolator with a minimum mean error (ME) and root mean square error (RMSE) [31, 32].

## Spatial scan statistical analysis

Using Kuldorff's SaTScan version 9.6 statistical software, the Bernoulli-based model was employed to detect the statistically significant spatial clusters with missing BPCR messages. A Bernoulli-based model was used in which events at particular places were analyzed, whether women received (Control) or missed (Case) BPCR messages. Scan statistics scanned the space gradually to determine the number of observed and expected observations inside the window at each cluster. The scanning window with the maximum likelihood was the most likely and high-performing cluster for being a case, and the level of significance was determined at a p-value <0.05.

## Spatial regression analysis

Spatial regression incorporates both global (ordinary list squares) and local(geographically weighted regression) analysis approaches [33, 34].

## Ordinary least squares (OLS) regression

The Ordinary Least Squares (OLS) model is a global statistical model used to evaluate and explain the relationship between outcome and explanatory variables [35]. It served as a tool for diagnosis as well as to pick the best predictors for the Geographically Weighted Regression (GWR) model. Findings from OLS regression are only reliable if the regression model meets all of the assumptions necessary by this method. In a properly fitted OLS model, coefficients of predictors should be statistically significant and have either a positive or negative sign. Furthermore, there should be no multicollinearity among explanatory variables [34, 36]. Multicollinearity was examined using the Variance Inflation Factor (VIF), and those variables with VIF > 10 were deemed as multicollinear (redundant) and were removed from the model. Furthermore, the Koenker Bp statistic had been utilized to determine whether the model could be used to do a spatially weighted regression analysis. The Koenker statistics were significant (p-value = 0.0198) in the current study, and GWR analysis was necessary to analyze the distribution of relevant predictors.

## Geographically weighted regression (GWR)

It is evident that a strong predictor in one cluster is not always a strong predictor in another. The application of GWR can identify this form of cluster variation (non-stationarity) of predictors [37]. Unlike OLS which fits a single linear regression equation to all of the data of all features (in this instance, the clusters), GWR uses data from neighboring features, so the GWR coefficient has distinct values for each cluster [38]. The six checks indicated for spatial regression analysis were also carried out [37]. Variables with p-values less than 0.05 are chosen and discussed based on their coefficients. Finally, the model with the lowest AICc score and the highest adjusted R-squared value was determined to be the ideal fit for the data.

## Ethical consideration and consent to participate

Written permission was received from ICF International to access both the DHS and GPS datasets following registration at the DHS Program office with possible justification. The accessed data were only used for the registered research and were not shared with anyone other than the co-authors. The DHS also declared that informed consent and assent was obtained from study participants or their legal guardian during the primary data collection [5]. Furthermore, the Institutional Review Board of Wachemo University College of Medicine and Health Sciences declared that no formal ethics approval was required in this case because it is secondary data, but to ensure the ethical compliance of the research process with the national and international standards.

## Results

### Socio-demographic characteristics of respondents

A total weighted sample of 4771 women was considered in the current study. The mean (±SD) age of women was 28.79(±6.61) years, with more than half (52.3%) of them belonging to the age group 25–34 years. One-third (33.7%) of study participants were from the Oromia region. The vast majority of respondents (81.7%) were from rural areas of the country, and more than half (54.1%) had no formal education. The proportion of women who missed BPCR messages was higher among those living in the poorest wealth quintile (51.0%), and received only one ANC visit(65.8%). Furthermore, 44.7% of women living in rural areas did not receive BPCR messages (Table 2).

### The level of missing BPCR messages

During the 2016 EDHS, 44.0% (95% CI: 42.6, 45.4%) of the surveyed women reported that they missed all BPCR messages during their last pregnancy. The proportion of women who missed the messages was highest in the Afar and Somali regions, with 78.1% and 67.2%, respectively. The top three messages not received by the majority of women were about arranging a potential blood donor (92.7%), identifying a companion during childbirth (90.8%), and securing emergency transportation (84.1%) (Table 3).

### Spatial autocorrelation of not receiving BPCR messages

The spatial autocorrelation analysis revealed that there was a considerable spatial variation in not receiving BPCR messages across the country (Global Moran's I value 0.218, p-value<0.001). The clustered patterns (on the right sides) suggest that missed BPCR messages happened at a high rate throughout the study area. The Z-score of 7.212 implies that there is less than a 1% possibility that this clustered pattern is due to random chance (Fig 1).

**Table 2. The weighted proportion of receipt of BPCR messages across selected characteristics of the respondents in Ethiopia, EDHS 2016.**

| Variable categories | Total [Weighted frequency (%)] | Receipt of BPCR message | | |
|---|---|---|---|---|
| | | Yes (%) | No (%) | Test statistics |
| **Age** | | | | |
| 15–24 | 1,236(25.9) | 670 (54.2) | 566(45.8) | $\chi^2$ = 9.12 |
| 25–34 | 2,496(52.3) | 1,377(55.2) | 1,119(44.8) | p = 0.071 |
| 35–49 | 1,039 (21.8) | 625(60.1) | 414(39.9) | |
| **Marital status** | | | | |
| Married and cohabiting | 4,481(93.8) | 2,507(56.0) | 1,974 (44.0) | $\chi^2$ = 2.02 |
| Not in marital relationship | 291(6.2) | 165 (56.7) | 126(43) | p = 0.089 |
| **Regions** | | | | |
| Tigray | 486(10.2) | 375(77.2) | 111(22.8) | $\chi^2$ = 423.70 |
| Afar | 37(0.8) | 8(21.9) | 29(78.1) | p<0.001 |
| Amhara | 1,104(23.1) | 727(65.9) | 377(34.1) | |
| Oromia | 1,607(33.7) | 744(46.3) | 863(53.7) | |
| Somali | 118(2.5) | 39(32.8) | 79(67.2) | |
| Benishangul | 56(1.2) | 27(48.9) | 29(51.1) | |
| SNNPR | 1,115(23.4) | 604(54.2) | 511(45.8) | |
| Gambella | 15(0.3) | 6(42.9) | 9(57.1) | |
| Harari | 13(0.3) | 8(60.8) | 5(39.2) | |
| Addis Ababa | 192(4.0) | 121(62.9) | 71(37.1) | |
| Diredawa | 29(0.6) | 13(45.6) | 16(54.4) | |
| **Residence** | | | | |
| Urban | 875(18.3) | 517(59.1) | 358 (40.9) | $\chi^2$ = 42.13 |
| Rural | 3,896 (81.7) | 2,155(55.3) | 1,741(44.7) | p = 0.011 |
| **Wealth index combined** | | | | |
| Poorest | 794(16.7) | 389(49.0) | 405(51.0) | $\chi^2$ = 94.16 |
| Poorer | 935(19.6) | 467 (49.9) | 468(50.1) | p<0.001 |
| Middle | 996(20.9) | 565(56.7) | 431(43.3) | |
| Richer | 967(20.2) | 568(58.8) | 398(41.2) | |
| Richest | 1,079(22.6) | 683(63.3) | 396(36.7) | |
| **Educational status** | | | | |
| No education | 2,580(54.1) | 1,395(54.1) | 1,185(45.9) | $\chi^2$ = 34.29 p<0.001 |
| Primary | 1,577(33.0) | 882(55.9) | 695(44.1) | |
| Secondary | 387(8.1) | 242(62.5) | 145(37.5) | |
| Higher | 227(4.8) | 153(67.5) | 74(32.5) | |
| **Family size** | | | | |
| >5 | 2,258(46.7) | 1,248(55.3) | 1,010(44.7) | $\chi^2$ = 2.43 p = 0.118 |
| ≤5 | 2,514(53.3) | 1,424(56.7) | 1,090(43.3) | |
| **Parity** | | | | |
| Nulliparous | 30(0.7) | 18(58.5) | 12(41.5) | $\chi^2$ = 0.897 |
| Primiparous | 1,193(28.9) | 691(58.9) | 502(42.1) | p = 0.217 |
| Multiparous | 2,241 (45.9) | 1,228(54.8) | 1,014(45.2) | |
| Grand multiparous | 1,307(27.5) | 736(56.3) | 571(43.7) | |
| **Frequency of ANC** | | | | |
| One visit | 313 (6.6) | 107(34.2) | 206(65.8) | $\chi^2$ = 191.19 |
| 2 visits | 581(12.2) | 258(44.4) | 323(55.6) | p<0.001 |
| 3 visits | 1,419(29.7) | 766(54.0) | 653(46.0) | |
| ≥4 visits | 2,458(51.5) | 1,541(62.7) | 918(37.3) | |

(*Continued*)

**Table 2.** (Continued)

| Variable categories | Total [Weighted frequency (%)] | Receipt of BPCR message | | |
|---|---|---|---|---|
| | | Yes (%) | No (%) | Test statistics |
| **Listen to radio** | | | | |
| Not at all | 3,162(65.5) | 1,749(55.3) | 1,413(44.7) | $\chi^2 = 69.13$ |
| Less than once a week | 793(16.9) | 451(56.9) | 342(43.1) | p<0.001 |
| At least once a week | 817(17.6) | 472(57.7) | 345(42.3) | |
| **Own mobile phone** | | | | |
| Yes | 1,166(26.7) | 712(61.1) | 454(38.9) | $\chi^2 = 54.19$ |
| No | 3,605(73.3) | 1,959 (54.3) | 1,646(45.7) | p<0.001 |
| **Covered by Health Insurance schemes** | | | | |
| Yes | 241(6.9) | 183(76.0) | 58(24.0) | $\chi^2 = 32.91$ |
| No | 4,530(93.1) | 2,488(54.9) | 2,042(45.1) | p<0.001 |
| **Access to money for seeking medical care** | | | | |
| Big problem | 2,560(53.7) | 1,354(52.9) | 1,207(47.1) | $\chi^2 = 46.34$ |
| Not a big problem | 2,211(46.3) | 1,318(59.6) | 893(40.4) | p<0.001 |

## The hot spot (Getis-Ord Gi*) analysis for missing BPCR messages

Hot spot analysis enables the detection of either high or low statistically significant coverage regions with the absence of BPCR messages. The northern, southern, and western Somali, central Afar, and Gambella regions were found as statistically significant hotspots (red dots) for not getting BPCR messages. While considerable cold spot areas were found in Tigray, Amhara, and Addis Ababa (Fig 2).

*Interpolation using GIS mapping.* Based on EDHS-2016 sampled data, areas with a relatively high proportion of missing BPCR messages (red) were predicted in Somali, Afar, and some parts of Oromia regions. In contrast, the predicted proportion of high receipt of BPCR messages (green shaded) encompasses the entire Tigray, Western Amhara, and central Addis Ababa regions (Fig 3).

*Spatial scan statistical (SaTScan) analysis.* The SaTScan spatial analysis detected a total of seven statistically significant SaTScan cluster areas with a high proportion of women missing

**Table 3. Disparities in the level of missing BPCR messages during pregnancy across regions of Ethiopia, EDHS 2016.**

| Regions | Women who missed each BPCR message [Weighted frequency (%)] | | | | | | |
|---|---|---|---|---|---|---|---|
| | About place of birth | About supplies needed for birth | About emergency transportation | About emergency fund/money | About companion during childbirth | About potential blood donors | Missing all messages |
| Tigray | 154(31.8) | 289(59.9) | 353(77.8) | 392(80.6) | 425(87.6) | 444(91.4) | 111(22.8) |
| Afar | 30(82.4) | 33(90.1) | 34(95.9) | 35(96.8) | 36(97.2) | 36(97.7) | 29(78.1) |
| Amhara | 426(38.6) | 845(76.6) | 960(88.0) | 965(87.4) | 998(90.4) | 1,005(91.0) | 377(34.1) |
| Oromia | 1,010(62.9) | 1,172(72.9) | 1,372(85.3) | 1,353(84.2) | 1,497(93.1) | 1,521(94.6) | 863(53.7) |
| Somali | 81(68.9) | 99(83.6) | 108(91.2) | 107(90.3) | 110(93.3) | 114(96.2) | 79(67.2) |
| Benishangul | 31(56.3) | 45(80.4) | 48(86.4) | 46(83.0) | 49(87.3) | 51(81.8) | 29(51.1) |
| SNNPR | 582(52.3) | 764(68.5) | 945(84.8) | 894(80.2) | 1,004(90.1) | 1,023(91.8) | 511(45.8) |
| Gambella | 9(61.1) | 11(75.6) | 13(83.8) | 13(87.3) | 14(93.0) | 14(93.0) | 9(57.1) |
| Harari | 7(52.6) | 11(79.6) | 10(78.6) | 11(82.9) | 12(87.2) | 12(88.6) | 5(39.2) |
| Addis Ababa | 111(58.0) | 103(53.6) | 142(74.1) | 134(68.8) | 163(85.2) | 177(92.4) | 71(37.1) |
| Diredawa | 18(60.9) | 23(79.9) | 26(89.6) | 24(83.2) | 28(94.8) | 28(94.4) | 16(54.4) |
| **Total** | 2,461(51.6) | 3,395(71.2) | 4,013(84.1) | 3,975(83.3) | 4,336(90.8) | 4,424(92.7) | 2,099(44.0) |

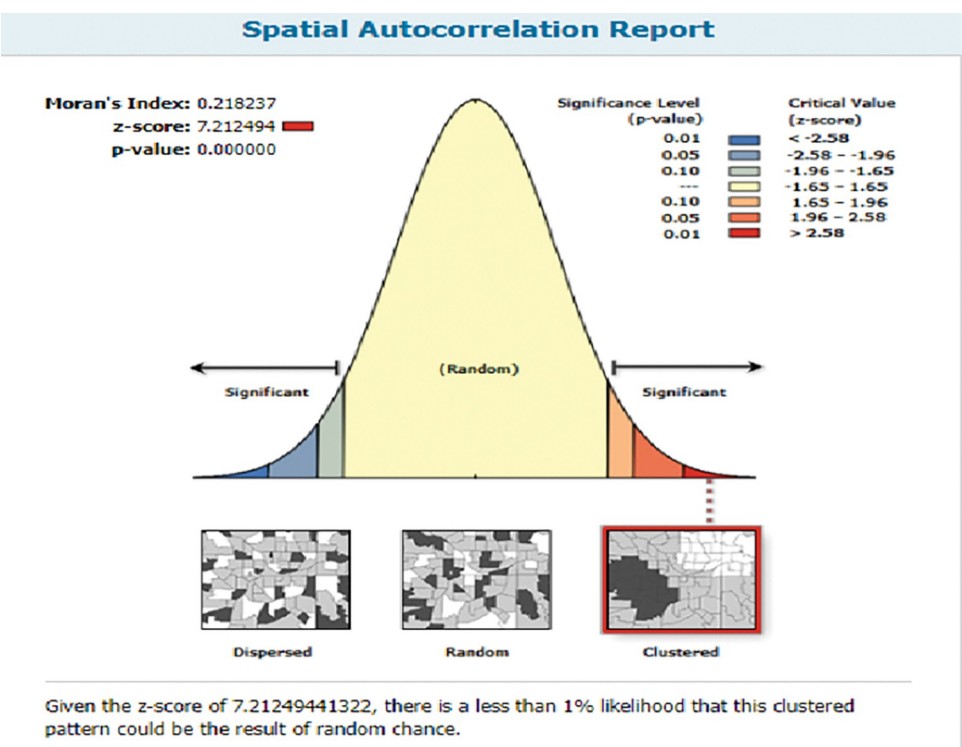

**Fig 1. The global spatial autocorrelation of missing BPCR messages during pregnancy in Ethiopia, EDHS 2016.**

BPCR messages, which means that the prevalence is higher inside the SaTScan circular window compared to outside the SaTScan window. The most likely primary SaTScan cluster of areas was detected in the Somali region at geographical coordinates of (6.745502 N, 44.259011 E) and with (LLR = 44.68, RR = 1.70, p < 0.001). This revealed that women within the spatial window had 1.70 times higher odds of missing BPCR messages as compared to their counterparts outside the spatial window (Table 4).

## Predictors of missing BPCR messages

**The global ordinary least square (OLS) analysis result.** OLS is a global regression model that uses a single equation to estimate the relationship between the dependent and independent variables, and it assumes the coefficients of each variable are homogenous across the study area. By meeting all of the fundamental assumptions, the initially employed spatial analysis, OLS model was fitted for the candidate explanatory variables. The model has been tuned to detect multicollinearity among the independent variables, with a mean VIF of less than 10. The adjusted R2 indicated that 24.2% (adjusted R2 = 0.242) of the variation in missing BPCR messages was explained by those six explanatory variables in the model. The robust probabilities for the intercept and all the included explanatory showed coefficient significance (p < 0.01) for the explanatory variables. Jarque-Bera statistics with p-values greater than 0.05 suggest that the model prediction using OLS was not biased (the assumption of residual normality was met). The Koenker statistics in the model, on the other hand, exhibited a statistically significant p-value (p = 0.0198), indicating that the regression model is inconsistent across the study area (as geographic position changes, so does the association of variables. This implies that the GWR model was deemed more suited for estimating model parameters. Consequently,

## Hot Spot and Cold Spot analysis of missing BPCR messages during pregnancy in Ethiopia

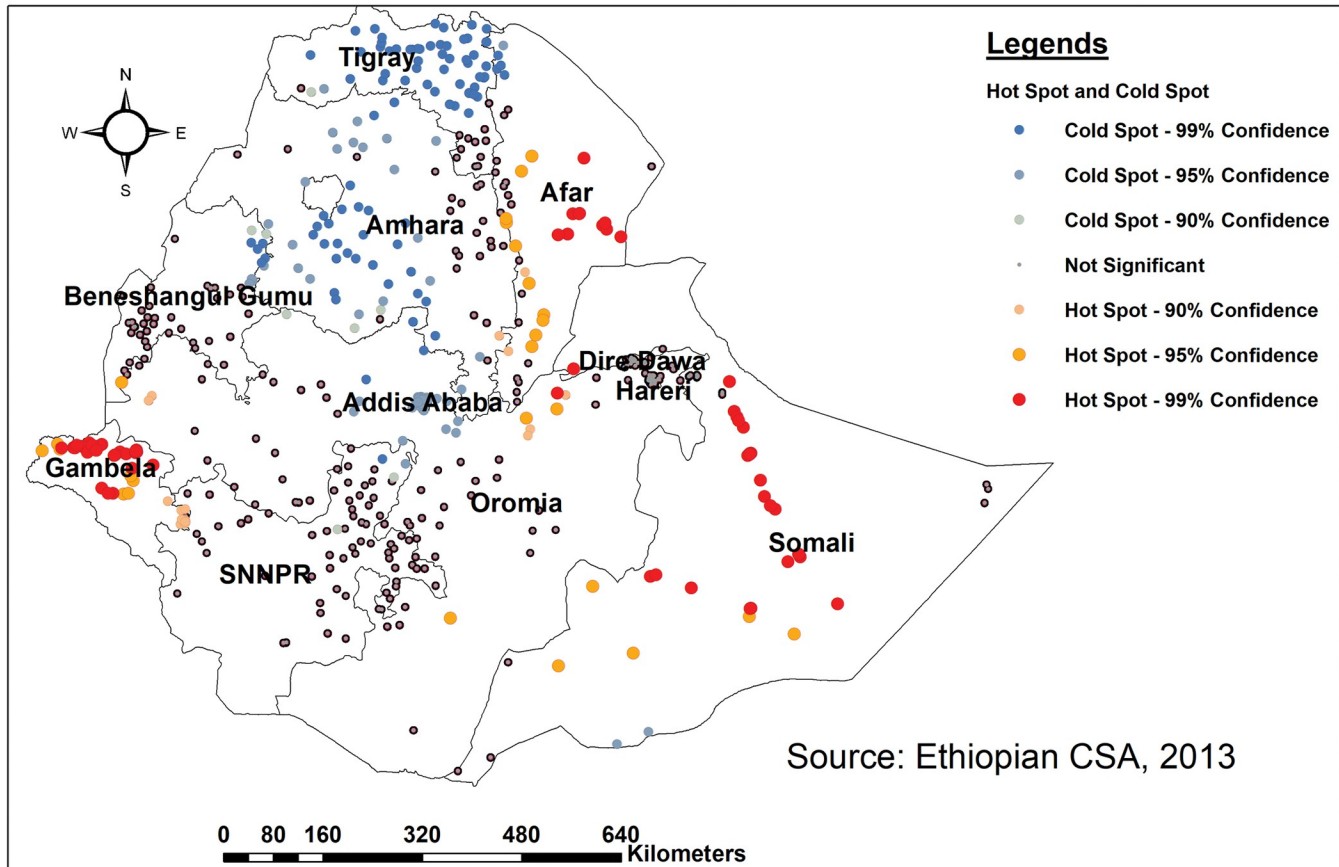

**Fig 2. Hot spot and Cold spot analysis of missing BPCR messages across regions in Ethiopia, EDHS 2016.**

being in the poorest wealth quintile, having only one ANC visit, having no access to listening to the radio, having difficulty accessing money to get medical care, not having a mobile phone, and not being covered by any health insurance scheme were the spatial determinants of hot spot areas for missing BPCR messages (Table 5).

### Geographically weighted regression (GWR)

Although the OLS model assumes that the association between each covariate and the outcome of interest is stationary across the study area, significant Koenker (BP) Statistics (p<0.01) show that this assumption is violated. To deal with this breach of the global (OLS) model's stationarity assumption, the local (GWR) model was fitted to provide credible estimates. The GWR analysis showed a considerable improvement over the global model (OLS). The AICc value decreased from 181.81 in the OLS model to 161.21 in the GWR model. Likewise, the adjusted $R^2$ in OLS ascended to 0.332 in GWR, indicating that the local model improved the capacity for predicting hotspots of missing BPCR messages. When the AICc values of two models (OLS versus GWR) differ by more than three, the model with the lower AICc is considered superior. Overall, GWR analysis outperformed the model developed with OLS in the current study.

## Interpolated spatial distribution of missing BPCR messages during pregnancy in Ethiopia

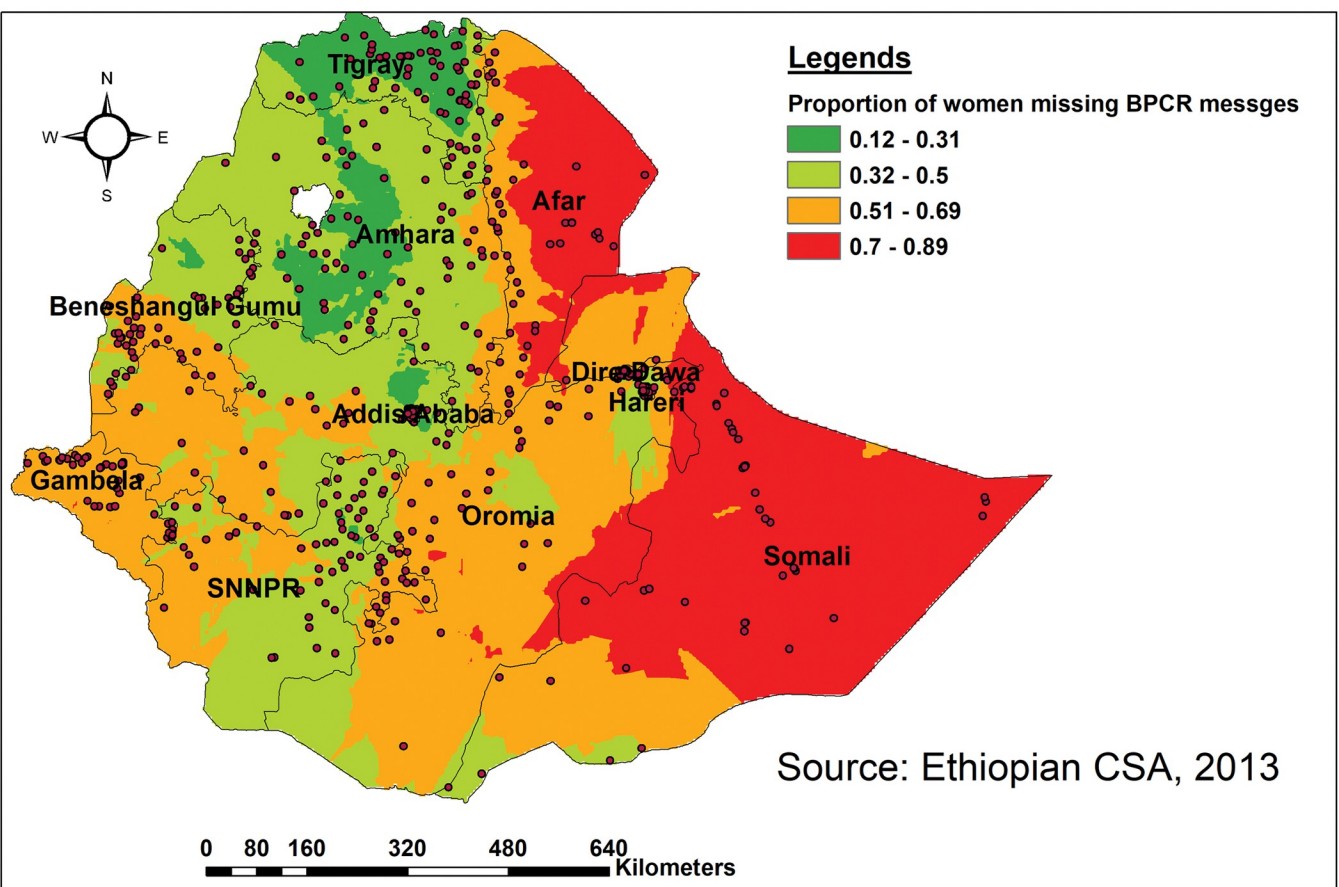

**Fig 3. Ordinary Kriging interpolation of the spatial distribution of missing BPCR messages in Ethiopia, EDHS 2016.**

The predicted coefficient of being in the poorest wealth quintile varies across the regions, revealing an uneven association between poverty and the proportion of missing BPCR messages. As the proportion of women living in the poorest wealth quintile increased, so did the proportion of missing BPCR messages in most parts of Afar and Somali, as well as in some parts of Gambella and Benishangul Gumuz. In those regions, the predicted coefficient ranges from 0.67 to 1.00 units. The northern part of the country, on the other hand, had the lowest coefficient for women in the poorest wealth quintile (northern, southern, and central Amhara region, and the majority of Tigray region). The red dotted area denotes a strong influence (high predicted coefficient) of the explanatory variables on missing BPCR messages (Fig 4).

Besides, receiving the lowest number of ANC visits (only one visit) was associated with the spatial variation of missing BPCR messages. As the proportion of women who received only one ANC visit increased, the percentage of missing BPCR messages among pregnant women in various parts of Afar, western, southern, and northern Somali, and some parts of SNNPR and Diredawa. The weakest relationship was detected in the majority of parts of Tigray, and Amhara regions (Fig 5).

The proportion of women who never listen to the radio was found to be strongly associated with an increased risk of missing BPCR messages, with the greatest effect observed in Somali (southern, northern, and western parts), Afar (eastern and southern parts), and Gambella

**Table 4. The most likely SaTScan clusters of areas with a high prevalence of missing BPCR messages in Ethiopia, EDHS 2016.**

| Most likely clusters | Enumeration areas (clusters) identified | Population | No. of case | Coordinates / Radius | Relative risk | LLR | P-Value |
|---|---|---|---|---|---|---|---|
| 1st most likely cluster | 490, 543, 92, 492, 171, 198, 146, 95, 85, 358, 164, 138, 497, 521, 588, 458, 553, 278, 214, 318, 573, 187, 239, 116, 568, 277, 527, 269, 556, 378, 630, 64, 439, 57, 480, 8, 210, 186, 454, 436 | 233 | 177 | (6.745502 N, 44.259011 E) / 360.39 km | 1.70 | 44.68 | <0.001 |
| 2nd most likely cluster | 75, 596, 632, 440, 178, 499, 334, 205, 570, 366, 427, 547, 276, 55, 368, 37, 135, 620, 389, 191, 571 | 127 | 100 | (11.382125 N, 41.702094 E) / 187.24 km | 1.73 | 28.78 | <0.001 |
| 3rd most likely cluster | 131, 16, 2, 250, 301, 323, 328, 356, 367, 379, 42, 48, 525, 530, 561, 625, 641, 9, 96, 618, 309, 435, 536, 370, 507, 592, 104, 233, 69, 426, 13, 603, 417, 346, 315, 284, 567, 343, 270, 105, 446, 106, 593, 219, 265, 221, 231, 549, 291, 469, 47, 643, 114, 63, 275, 193, 395, 448, 337, 508, 165, 462, 285, 326, 248, 581, 317, 203, 175, 374, 6, 197, 416, 526, 17, 46, 595, 371, 465, 552, 168, 243, 459, 409, 433, 299, 554, 407, 119, 563, 324, 437, 209, 325, 457, 376, 335, 558, 569, 177, 477, 555, 65, 124, 304, 621, 256, 349, 586, 207, 88, 154 | 828 | 479 | (0.000000 N, 0.000000 E) / 4096.34 km | 1.32 | 26.77 | <0.001 |
| 4th most likely cluster | 362, 127, 235, 263 | 37 | 34 | (13.889667 N, 39.944065 E) / 16.79 km | 2.00 | 17.73 | <0.001 |
| 5th most likely cluster | 518, 405, 576, 468, 313, 365 | 47 | 39 | (6.743022 N, 38.962164 E) / 41.43 km | 1.81 | 13.67 | <0.001 |
| 6th most likely cluster | 564, 39, 230, 51, 484 | 31 | 28 | (9.555410 N, 40.326165 E) / 36.96 km | 1.96 | 13.63 | <0.001 |
| 7th most likely cluster | 372, 93, 412, 476, 506, 453, 491, 441, 557, 594, 30, 25, 166, 380, 43, 74, 282, 644, 151, 273, 631, 111, 535, 519, 546, 613, 471, 190, 5, 607, 467, 363, 473, 606, 202, 101, 185, 115, 390, 513, 444, 140, 311, 173, 385, 27, 352, 224, 514, 493, 443 | 364 | 209 | (8.949350 N, 41.312402 E) / 95.86 km | 1.26 | 9.73 | 0.025 |

(northern and southern parts). The weakest association with the predicted coefficient of 0.04 to 0.32 was found in the northern Tigray and some parts of the Amhara regions (Fig 6).

Furthermore, not having a mobile phone was associated with a regional difference in not getting BPCR messages. Women in the northern, southern, and western parts of the Somali region and the eastern border of Afar who did not have mobile phones had a higher likelihood of missing BPCR messages (Fig 7).

**Table 5. Summary of OLS results for not receiving BPCR messages in Ethiopia, EDHS 2016.**

| Variable | Coefficient | SE | t-Statistic | Probability | Robust SE | Robust t-statistics | Robust probability | VIF |
|---|---|---|---|---|---|---|---|---|
| Intercept | -0.045 | 0.084 | -4.21 | 0.021 | 0.067 | -3.32 | <0.001 | — |
| Women in the poorest wealth quintiles | 0.233 | 0.037 | 6.33 | <0.001 | 0.041 | 5.65 | <0.001 | 1.39 |
| Women who received only one ANC visit | 0.287 | 0.079 | 3.62 | <0.001 | 0.070 | 4.11 | <0.001 | 1.17 |
| Women who never listened to a radio | 0.144 | 0.043 | 3.32 | <0.001 | 0.048 | 2.99 | 0.003 | 1.47 |
| Women who didn't own a mobile phone | -0.132 | 0.035 | -3.75 | <0.001 | 0.036 | -3.60 | <0.001 | 1.26 |
| Women who faced difficulty in getting money | 0.134 | 0.038 | 3.51 | <0.001 | 0.041 | 3.26 | 0.001 | 1.25 |
| Women who were not covered by any health insurance schemes | 0.367 | 0.085 | 4.32 | <0.001 | 0.066 | 5.51 | <0.001 | 1.03 |
| **OLS Diagnostics** | | | | | | | | |
| Number of Observations: | 607 | | Akaike's Information Criterion (AICc) | | | | 181.81 | |
| Multiple R-Squared | 0.252 | | Adjusted R-Squared [d]: | | | | 0. 242 | |
| Joint F-Statistic | 28.771 | | Prob(>F), (7,599) degrees of freedom | | | | <0.001 | |
| Joint Wald Statistic | 253.005 | | Prob(>chi-squared),(7) degrees of freedom | | | | <0.001 | |
| Koenker (BP) Statistic | 3.959 | | Prob(>chi-squared), (7) degrees of freedom | | | | 0.0198 | |
| Jarque-Bera Statistic | 1.214 | | Prob(>chi-squared), (2) degrees of freedom | | | | 0.0604 | |

## Proportion of women living in the poorest wealth quintile

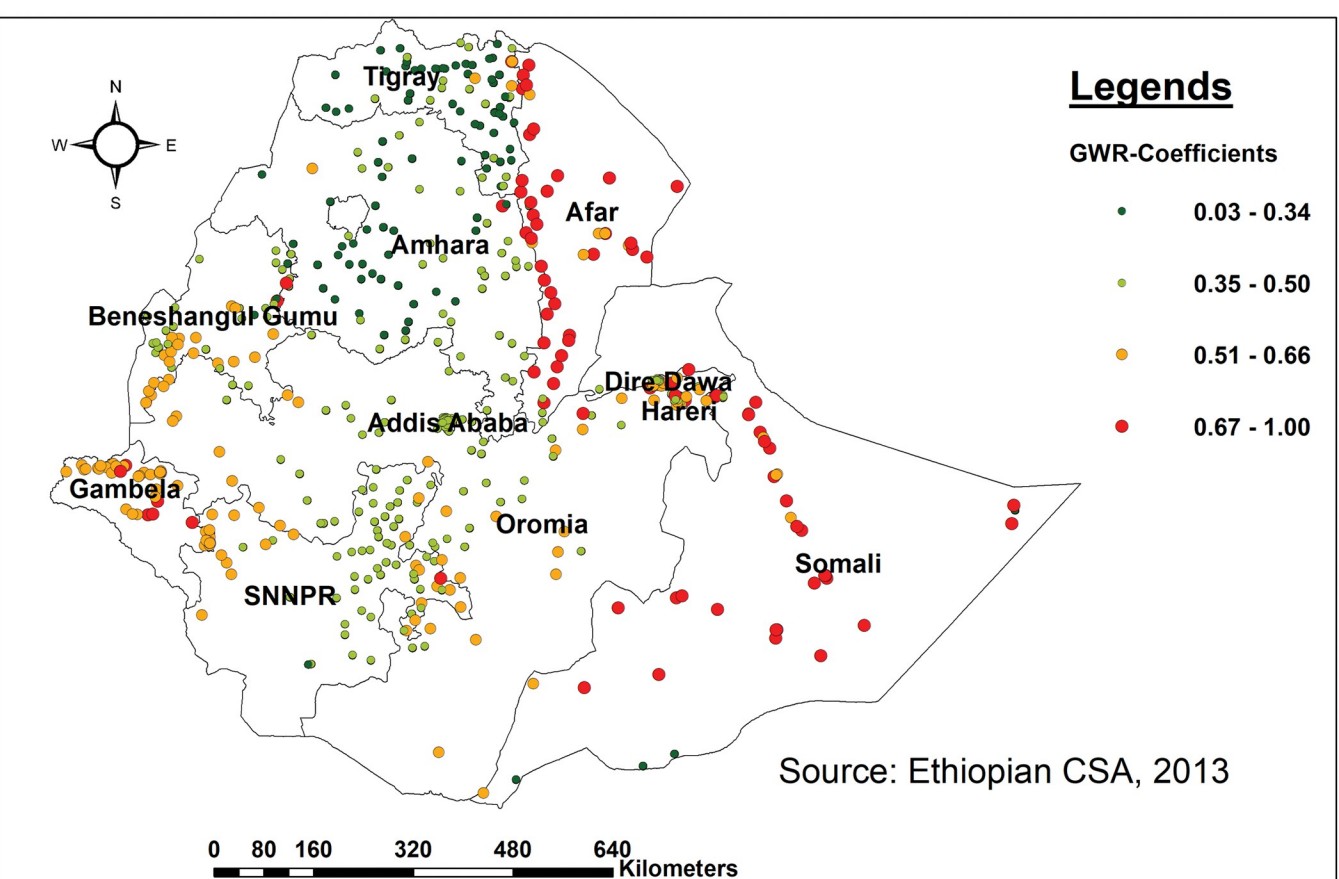

**Fig 4. Spatial mapping of GWR coefficients of women in the poorest wealth quintile for predicting missing BPCR messages in Ethiopia, EDHS 2016.**

The predicted coefficient of difficulty in obtaining money for medical care on the level of missing BPCR messages varies by geographical region. As the proportion of women who faced difficulty in getting money increased, the failure to receive BPCR messages in the northern and southern portions of Somali, as well as the central and south-western parts of the Afar regions. Additionally, it encompasses some parts of Diredawa and Hareri, Oromia (southern), Gambella (northern), and the Benishangul Gumuz (southern) regions (Fig 8).

Finally, not covered by any type of health insurance scheme has a positive relationship with missing BPCR messages. As the proportion of women who were not covered by any HI schemes increased, the prevalence of missing BPCR increased greatly in all regions except Amhara and Tigray (Fig 9).

## Discussion

The overall prevalence of missing BPCR messages in Ethiopia was found to be 44.0% (95% CI: 42.6, 45.4). Because there was a dearth of studies on the topic at the national level, it was difficult to compare the prevalence to other findings. The findings of spatial global Moran's analysis found that the proportion of women who missed BPCR messages significantly varied across regions.

## Proportion of women who received only one ANC visit

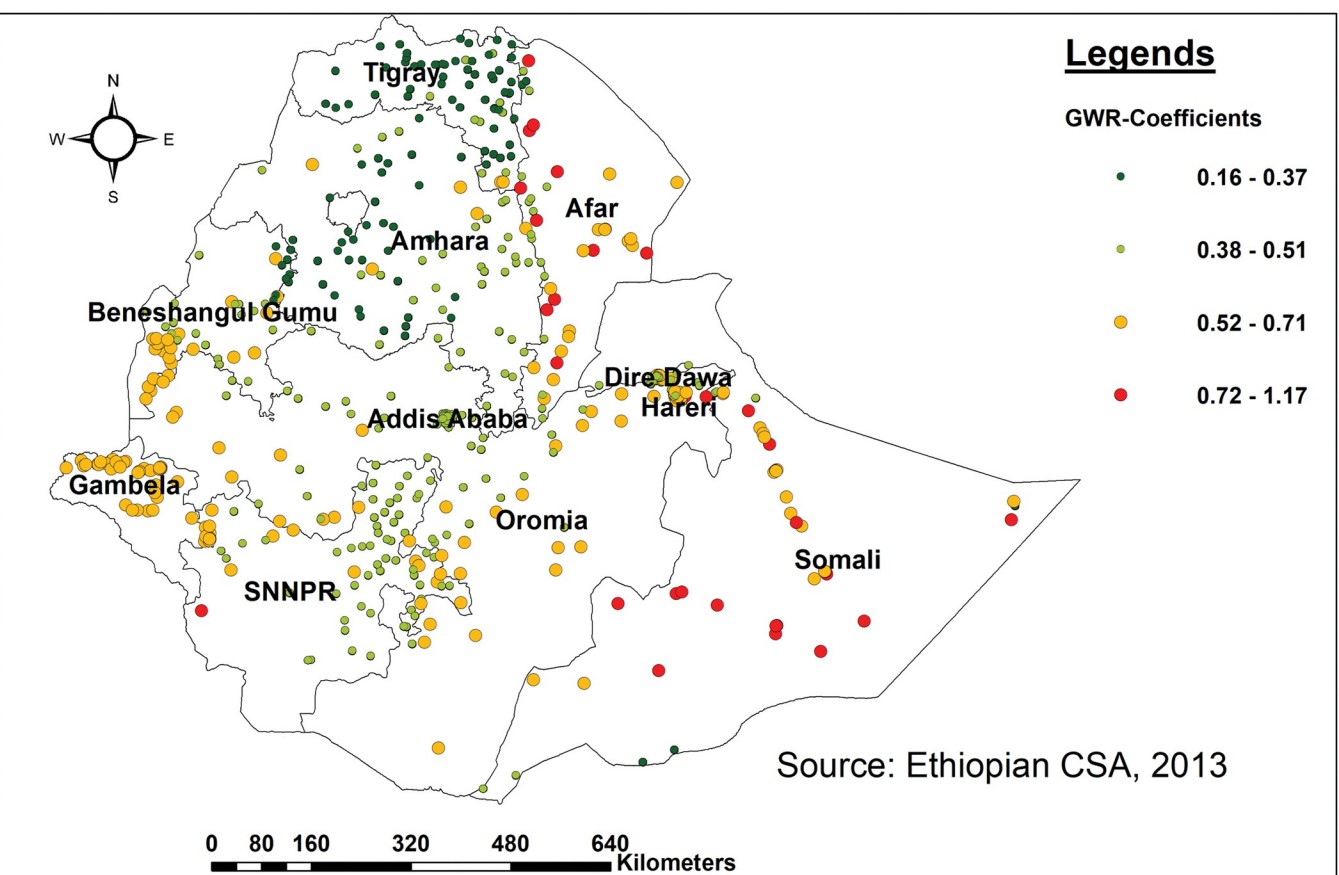

**Fig 5. Spatial mapping of GWR coefficients of women with only one ANC visit for predicting missing BPCR messages in Ethiopia, EDHS 2016.**

As per hot spot analysis, the vast majority of Somali, central Afar, and Gambella regions had a high clustering (hot spot) of not getting BPCR messages. Those regions were known for low uptake of maternal and child health services [39–41]. Prior small-scale studies conducted in those regions also supported this finding [42–44]. This clustering might be due to a variety of factors. To begin, those two regions are known to have fewer healthcare facilities, inadequate healthcare providers, and lesser availability of medical supplies and equipment to access maternal health services when compared to other regions [40, 45]. In addition, these regions frequently have remote and difficult-to-reach areas with limited transportation, cultural practices that may take precedence over modern medical care, and lower socioeconomic conditions, such as poverty and a lack of education, making it difficult for pregnant women to access healthcare services on time [40, 41, 46]. Furthermore, some populations in these regions live nomadic or semi-nomadic lives, which might make it difficult for pregnant women to get consistent healthcare services because their locations shift over time [40, 47]. All these can lead to a lack of proper follow-up during pregnancy which might lead to missing BPCR messages.

As per the local (GWR) model, being in the poorest wealth quintile, receiving only one ANC visit, never listening to the radio, facing difficulty in accessing money, not having a mobile phone, and not being covered by any health insurance scheme were identified as significant spatial predictors of missing BPCR messages in.

## Proportion of women who never listen to the radio

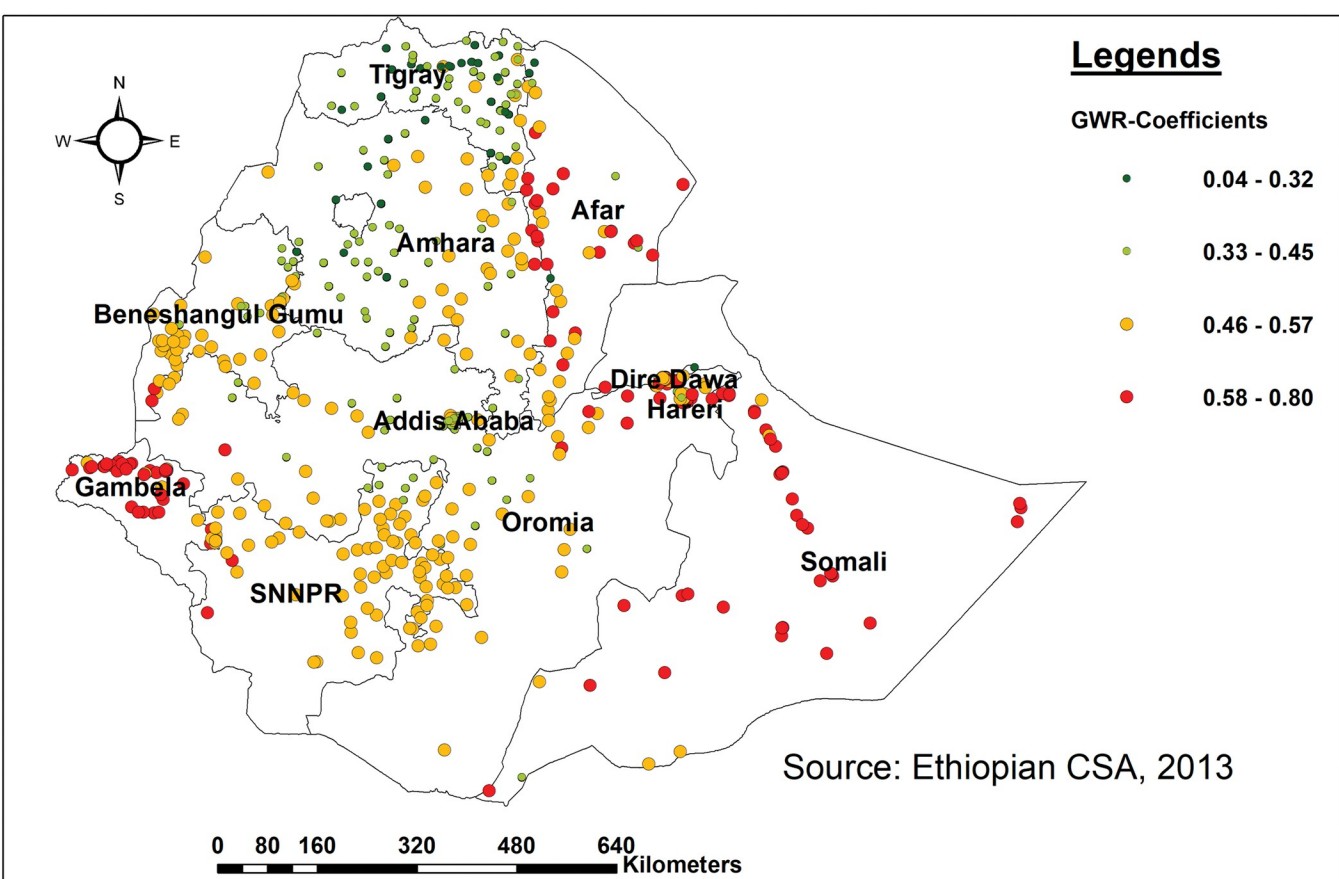

**Fig 6. Spatial mapping of GWR coefficients of women who never listen to the radio for predicting missing BPCR messages in Ethiopia, EDHS 2016.**

Living in a household with the poorest wealth quintile was identified as a significant predictor of hot spot for not receiving BPCR messages in most parts of Afar and Somali, as well as in some parts of Gambella and Benishangul Gumuz. In those regions, the predicted coefficient ranges from 0.67 to 1.00 units. This was supported by studies conducted in Nigeria [48], Ghana [49], and Ethiopia [14, 50, 51]. This association may be explained by the low utilization of maternal health services in those regions [39–41]. In addition, those women in the poorest wealth quintile are more likely to experience financial barriers in accessing maternal healthcare especially ANC, which is a vital entry point to get information about BPCR messages. In addition, poverty and economic challenges can cause extra pressures on women and families, by hampering access to media and the ability to make decisions [52]. All of this might influence their capacity to seek healthcare services as well as receive key BPCR messages. It is vital to emphasize that being in the poorest wealth quintile might lead to a cycle of inadequate service use and poor health outcomes. Improving the issue necessitates multifaceted approaches that address economic disparities, enhance public infrastructures, promote education, and engage communities in addressing cultural norms and beliefs [53].

Similarly, the frequency of ANC visits is a key spatial predictor of hotspots of missing BPCR messages across the region. The likelihood of missing BPCR messages increased in the vast majority of Afar, western, southern, and northern Somali, and some parts of SNNPR and

## Proportion of women who hadn't own mobile phone

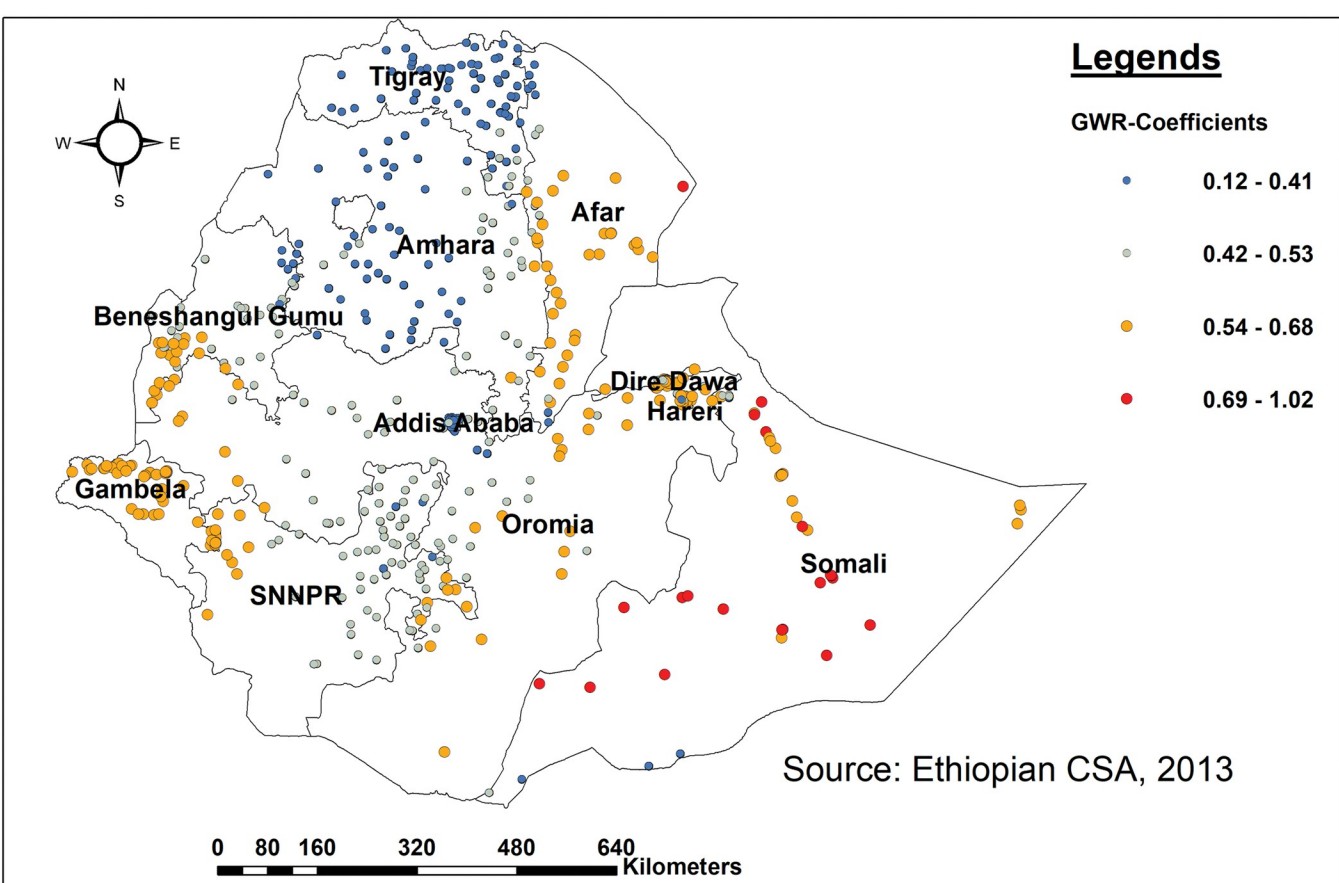

**Fig 7. Spatial mapping of GWR coefficients of women who didn't own a mobile phone for predicting missing BPCR messages in Ethiopia, EDHS 2016.**

Diredawa as the proportion of women who received only one ANC visit increased. Studies conducted in Tanzania [54], Egypt [55], Kenya [56], and Ethiopia [43, 51] found that having a low number of ANC visits was positively related to not receiving BPCR messages. This could be attributed to a variety of reasons. To begin, receiving BPCR messages entails a sequence of steps that must be followed to ensure a safe pregnancy and childbirth, and if a woman just attends one visit, she may not obtain comprehensive information about all of these elements. In addition, with changing circumstances, not getting adequate visits resulted in missing out on all of the WHO-recommended BPCR messages. Overall, because pregnancy is a dynamic process, obtaining only one ANC visit can result in missing BPCR messages due to a lack of comprehensive education, support, tailored updates, engagement, and follow-ups that subsequent visits offer [57, 58]. Thus, pregnant women need to attend all directed ANC visits to ensure they obtain the appropriate information and care for a safe pregnancy and childbirth.

Failure to listen to the radio was found to be strongly associated with an increased risk of missing BPCR messages in Somali (southern, northern, and western parts), Afar (eastern and southern parts), and Gambella (northern and southern parts). This was supported by studies conducted in Uganda [59], and Ethiopia [12, 60, 61]. As a result of not tuning in to the radio, women may remain unaware of and miss out on services available to them throughout pregnancy and childbirth, becoming alienated or feeling more lonely and less supported during

## Proportion of women who faced difficulty of getting money to access mediacl care

**Fig 8. Spatial mapping of GWR coefficients of women who faced difficulty in getting money for predicting missing BPCR messages in Ethiopia, EDHS 2016.**

their pregnancy experience [62]. All these could contribute to missing BPCR messages during pregnancy. Subsequently, a lack of radio access can limit women's ability to make informed decisions and contribute to discrepancies in the uptake of key BPCR messages. Hence, a concerted effort is needed to enhance the accessibility of radio, to ensure that no woman is deprived of those key BPCR messages during pregnancy. Similarly, not having a mobile phone was associated with a regional difference in not getting BPCR messages in the northern, southern, and western parts of the Somali region and the eastern border of Afar. Studies conducted elsewhere [63–65] showed that there was a strong association between having a mobile phone and receipt of BPCR messages. This could be because women without mobile phones were less likely to receive health-related information, appointment reminders, health awareness campaigns, and peer support, resulting in BPCR messages being overlooked [66]. Overall, mobile phones provide a versatile and accessible platform for bridging gaps in maternal healthcare service usage through providing information, support, reminders, and access to healthcare providers, particularly in areas with limited healthcare infrastructure [67]. As a result, it is vital to prioritize digital literacy, network coverage, and cost to promote fair access for all women.

Finally, being not covered by health insurance schemes depicts a positive link with missing BPCR messages; as the proportion of women who were uninsured increased, the prevalence of missing BPCR increased significantly in all regions except Amhara and Tigray. Studies conducted in Rwanda [68], and Nigeria [69] showed high uptake of BPCR messages among those

## Proportion of women who were not covered by any Health Insurance scehemes

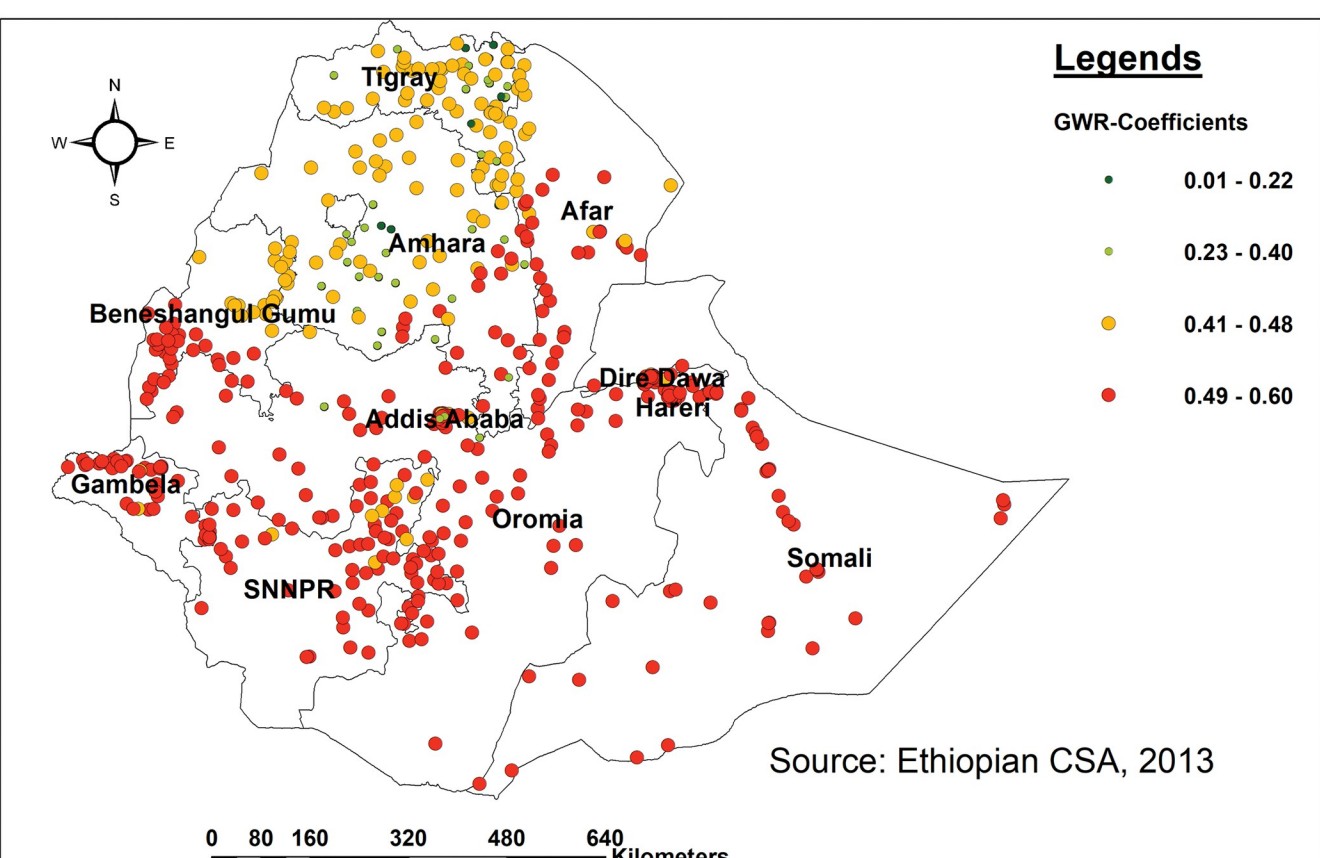

**Fig 9. Spatial mapping of GWR coefficients of women who were not covered by health insurance schemes for predicting missing BPCR messages in Ethiopia, EDHS 2016.**

women who were insured. This could be because a lack of health insurance coverage can create hurdles to access and continuity of care, as well as engagement with health service delivery points, due to cost concerns and financial hardship [70, 71]. As a result, people may connect with healthcare providers less frequently, missing out on vital information about BPCR messages. On the other hand, women who were covered by health insurance were more likely to visit health facilities, and this trend made them more familiar with the healthcare system, which in turn boosted their health-seeking behavior during pregnancy [72].

The study has the following strengths. First, the findings from the current study were based on an analysis of nationally representative data, making the findings more generalizable. Also, the findings from the Hot Spot, SaTScan, and GWR analyses can assist government and program planners in designing geographically focused public health interventions to boost BPCR message delivery. Despite the strengths, the findings should be interpreted considering the following limitations. Firstly, to ensure the privacy of respondents, the geographical coordinates of clusters were displaced by up to 2km in urban areas, 5km for most clusters in rural areas, and 10km for 1% of clusters in rural areas; this may impact estimated cluster effects in the spatial regression. In addition, due to the cross-sectional nature of the data, the findings may be affected by recall- and social desirability biases and it is difficult to establish a temporal/causal relationship between the outcome and explanatory variables.

## Conclusion

The level of missing BPCR messages during pregnancy was found to be high in Ethiopia with a significant spatial variation across regions. Living in the poorest wealth quintile, receiving only one ANC visit, not having access to listening to the radio, having difficulty accessing money to get medical care, not having a mobile phone, and being not covered by health insurance were identified as significant predictors of hot spots for missing BPCR messages. The level of missing BPCR messages during pregnancy was found to be high in Ethiopia, with significant local variation. As a result, policymakers at the national level and local planners should develop strategies and initiatives that enhance women's economic capacities, health-seeking behavior, and media exposure. Furthermore, the regional authorities should focus on strategies that promote universal health coverage through enrolling citizens in health insurance schemes.

## Supporting information

**S1 File. Output for incremental spatial autocorrelation.**
(PDF)

## Acknowledgments

We are grateful to ICF macro (Calverton, USA) for providing the 2016 DHS data of Ethiopia.

## Author Contributions

**Conceptualization:** Aklilu Habte.

**Data curation:** Aklilu Habte.

**Formal analysis:** Aklilu Habte.

**Methodology:** Aklilu Habte, Samuel Hailegebreal, Tamirat Melis, Dereje Haile.

**Project administration:** Aklilu Habte.

**Software:** Aklilu Habte, Samuel Hailegebreal.

**Validation:** Aklilu Habte.

**Visualization:** Aklilu Habte.

**Writing – original draft:** Aklilu Habte, Samuel Hailegebreal.

**Writing – review & editing:** Aklilu Habte, Tamirat Melis, Dereje Haile.

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
