## [Decision Letter · Decision Letter 0]

2 Oct 2023

PONE-D-23-28170Modeling spatial predictors of missing birth preparedness and complication readiness (BPCR) messages during pregnancy in Ethiopia: a Geographically Weighted RegressionPLOS ONE

Dear Dr. Habte,

Thank you for submitting your manuscript to PLOS ONE. After careful consideration, we feel that it has merit but does not fully meet PLOS ONE’s publication criteria as it currently stands. Therefore, we invite you to submit a revised version of the manuscript that addresses the points raised during the review process.

We look forward to receiving your revised manuscript.

Kind regards,

Kanchan Thapa, MPH, MPhil

Academic Editor

PLOS ONE

Journal Requirements:

3. We note that Figures 2-9 in your submission contain map images which may be copyrighted. All PLOS content is published under the Creative Commons Attribution License (CC BY 4.0), which means that the manuscript, images, and Supporting Information files will be freely available online, and any third party is permitted to access, download, copy, distribute, and use these materials in any way, even commercially, with proper attribution. For these reasons, we cannot publish previously copyrighted maps or satellite images created using proprietary data, such as Google software (Google Maps, Street View, and Earth). For more information, see our copyright guidelines: http://journals.plos.org/plosone/s/licenses-and-copyright.

1.) You may seek permission from the original copyright holder of Figures 2-9 to publish the content specifically under the CC BY 4.0 license.  

2.) If you are unable to obtain permission from the original copyright holder to publish these figures under the CC BY 4.0 license or if the copyright holder’s requirements are incompatible with the CC BY 4.0 license, please either i) remove the figure or ii) supply a replacement figure that complies with the CC BY 4.0 license. Please check copyright information on all replacement figures and update the figure caption with source information. If applicable, please specify in the figure caption text when a figure is similar but not identical to the original image and is therefore for illustrative purposes only.

**Additional Editor Comments:**

In Abstract section- I found it should be rewritten. First some statement should be given why birth preparedness is important for which generally we start with some scenario of maternal or new-born mortality in global, regional and national picture. Also you can highlight same thing in the background section of the main text. Work on your abstract section to make it more attractive.

I understand that Ethiopia might not have prior spatial analysis of BPP however please use soft word like there is limited evidence of prior publication of such article or something in more soft way. Since lack of publication does not indicate lack of evidence. There might have some policy or programmatic document which might not be reviewed by authors. Thus, I suggest to rewrite the whole abstract once.

You have mentioned about geographic weighted regression in the title which seems to be like you want to attract more readers based on type of analysis you have performed in the paper. Does the word (GWR) title have any scientific values to be mentioned in the title of the paper? Please correct accordingly.

Introduction

Please give general background of MMR, NMR of the global scale rather the stating despite of the change by % etc. Please rewrite the background section and link why the spatial analysis is important. You have written something about birth preparedness message nicely however I think still there are opportunities to concise and make it more attractive and relevant.

Methods

Also mention about how weighting been done. If possible provide stata syntax as supplementary file. Adhere and improve critically based on reviewer comments too.

Results

Formatting of tables and figure should be done as per the PLOS one requirement. Please review one published paper so that you can work accordingly. Analytical interpretation of the findings has been lacking throughout the text. Please work on it. Also, take care of table and figure headings and presentation.

Discussion

Is this really the first study published from Ethiopia? You have mentioned about the BPCR message prevalence, I suggest these should be mentioned on your background section too. First paragraph is quite long, I suggest you to focus on to discuss about summary of your findings in this paragraph.

Conclusion

Make your conclusion catchier. Don’t copy information from result section. Moreover, some of the information fits into the recommendation section in the discussion section.

I suggest you to review the comments attached below for further improvement of the paper. Please review and revise the paper critically.  Please ensure that your decision is justified on PLOS ONE’s publication criteria and not, for example, on novelty or perceived impact.

Reviewers' comments:

Reviewer's Responses to Questions

**Comments to the Author**

1. Is the manuscript technically sound, and do the data support the conclusions?

Reviewer #1: Partly

Reviewer #2: Yes

2. Has the statistical analysis been performed appropriately and rigorously? 

Reviewer #1: I Don't Know

Reviewer #2: Yes

3. Have the authors made all data underlying the findings in their manuscript fully available?

Reviewer #1: No

Reviewer #2: Yes

4. Is the manuscript presented in an intelligible fashion and written in standard English?

Reviewer #1: No

Reviewer #2: Yes

5. Review Comments to the Author

Reviewer #1: Dear Author,

I enjoyed reading your paper however there are some room for the improvement. I have commented on the PDF version of your paper and highlighted the main part. However some others things need to be improved aside of the highlighted parts as follow:

1. Please review the Plos One guideline intensively.

2.Please proof read thoroughly by each and every team members

3. Polish the language

4. Revise the formatting

5. Reorganized some of the paragraph that I have highlighted in the Pdf file.

Thank you so much!

Reviewer #2: This is an interesting paper and the research question is of great practical significance. The following are some suggestions for revision.

Q1: As far as I know, OLS (Ordinary Least Squares) is most suitable for situations where the dependent variable is a continuous variable. In this paper, it seems that the OLS model was used for modeling the relationship between "missing BPCR messages" and explanatory variables, whereas the "missing BPCR messages" is a binary variable. Please provide the rationality of adopting this method.

Q2: There are some spelling errors in the paper. For example, "Ordinary List Squares" should be replaced by "Ordinary Least Squares". In table 2, the test statistic "X2" should be replaced by Greek letter χ, and the "2" would be marked as superscript.

Q3: In table 2, there are some calculation errors. For example, "126(43)" should be replaced by "126(43.3)"; In Somali, the total frequency is 11, whereas the sum of "Receipt of BPCR message" and "Not receipt of BPCR message" was inconsistent with the total. So please double check the accuracy of each figure in the table.

6. PLOS authors have the option to publish the peer review history of their article (what does this mean?). If published, this will include your full peer review and any attached files.

Reviewer #1: **Yes: **Rita Adhikari

Reviewer #2: No

---

## [Author Response · Author response to Decision Letter 0]

15 Nov 2023

A point-by-point response to editor and reviewers

Authors’ Response to Academic Editor

Dear: Kanchan Thapa, MPH, MPhil, Academic Editor, Plos One

We thank you for a thorough reading and constructive comments and suggestions on our manuscript and for the opportunity to revise and resubmit. We are pleased to submit the revised version of the manuscript titled “Spatial variation and predictors of missing birth preparedness and complication readiness (BPCR) messages in Ethiopia: Spatial analyses” for your consideration in the special collection of Plos One. The comments of the editors and the reviewers were highly insightful and enabled us to greatly improve the quality of our manuscript. In this revised manuscript we made substantial changes to address your concerns in a point-by-point response. We appreciate your time and look forward to your response and we are very keen to incorporate further comments, if any, for the betterment of the final manuscript.

On the following pages, you will find our responses to the comments and suggestions raised by the esteemed editor and reviewers. 

Sincerely, 

Aklilu Habte(MPH)(corresponding author)

aklilihabte57@gmail.com

Response to Journal requirements

Response: we already prepared the manuscript as per the journal requirement and again we rechecked the compliance towards it during the submission of our revised manuscript.

2. We note that you have indicated that data from this study are available upon request. PLOS only allows data to be available upon request if there are legal or ethical restrictions on sharing data publicly. In your revised cover letter, please address the following prompts:

Response: as we mentioned in the ‘data availability’ section of the initial manuscript, the data supporting the findings of this study can be obtained in anonymized form from the Demographic and Health Survey website at https://www.dhsprogram.com upon reasonable request in the same manner as the authors. So, now, as per your suggestion, we incorporate the statement in the ‘Cover letter’ of the "Revised Manuscript with Track Changes"

3. If there are no restrictions, please upload the minimal anonymized data set necessary to replicate your study findings as either Supporting Information files or to a stable, public repository and provide us with the relevant URLs, DOIs, or accession numbers.

Response: as per the DHS office rules and regulations, it is impossible to share the data with third parties other than the authors, and the data can be accessed at https://dhsprogram.com/data/dataset_admin/index.cfm based on a reasonable request.

4. We note that Figures 2-9 in your submission contain map images which may be copyrighted. All PLOS content is published under the Creative Commons Attribution License (CC BY 4.0), which means that the manuscript, images, and Supporting Information files will be freely available online, and any third party is permitted to access, download, copy, distribute, and use these materials in any way, even commercially, with proper attribution. For these reasons, we cannot publish previously copyrighted maps or satellite images created using proprietary data, such as Google software (Google Maps, Street View, and Earth). 

You may seek permission from the original copyright holder of Figures 2-9 to publish the content specifically under the CC BY 4.0 license. 

Response: We appreciate your concern to assure the ethical issues. However, all the aforementioned figures(2-9) in our manuscript are not copyrighted rather they are the result of spatial analysis that we have run in ArcGIS and SaTScan software. The GPS and DHS data that contain Shapefile and other relevant variables were obtained from the DHS office by explaining the objective of the study through online requests. Then, in order to get those figures, we import the relevant data extracted from the 2016 Ethiopian Demographic Health Survey reports and the shapefile of Ethiopia obtained from the 2016 Ethiopian Central Statistical Agency (CSA).To indicate this, we already cited the source of the shapefile alongside each figure. The shape file that we used to construct the figures can be accessed by one of the following links:

https://data.humdata.org/dataset/cb58fa1f-687d-4cac-81a7-655ab1efb2d0 .

https://gadm.org/download_country.html

Therefore, the maps presented in our study are not copyrighted rather they were the outputs of our spatial analysis results which are the result of those Shapefiles and projected CVS files in ArcGIS. This is the actual procedure that we employed in our present and earlier studies, as well as other Ethiopian researchers. Again we assure you that the figures presented in our study are not copyrighted but rather our spatial analysis results.

Response to Additional Editor Comments: 

Comment 1: In the Abstract section- I found it should be rewritten. First some statement should be given why birth preparedness is important for which generally we start with some scenario of maternal or new-born mortality in global, regional and national picture. Also you can highlight same thing in the background section of the main text. Work on your abstract section to make it more attractive. I understand that Ethiopia might not have prior spatial analysis of BPP however please use soft word like there is limited evidence of prior publication of such article or something in more soft way. Since lack of publication does not indicate lack of evidence. There might have some policy or programmatic document which might not be reviewed by authors. Thus, I suggest to rewrite the whole abstract once.

Response: Thank you for your insightful suggestion and comment on the whole abstract. We get all of them as valuable input to improve the readability of our work, and accordingly, we have made some amendments and all of them were highlighted in the ‘Abstract’ section of the "Revised Manuscript with Track Changes" Page 2

Comment 2: You have mentioned about geographic weighted regression in the title which seems to be like you want to attract more readers based on type of analysis you have performed in the paper. Does the word (GWR) title have any scientific values to be mentioned in the title of the paper? Please correct accordingly.

Response: Thank you for the comment. We entirely accept your suggestion and we replaced it with ‘spatial analyses’. The corrected version is highlighted in the ‘title’ of the "Revised Manuscript with Track Changes" Page 1, Line 1-3

Comment 3: in the introduction section, please give a general background of MMR, NMR of the global scale rather the stating despite of the change by % etc. Please rewrite the background section and link why the spatial analysis is important. You have written something about birth preparedness message nicely however I think still there are opportunities to concise and make it more attractive and relevant.

Response: Thank you for your insightful suggestions. The revised statement was highlighted in the ‘Introduction’ section of the "Revised Manuscript with Track Changes" on Page 3, Lines 64-66 and 69-71. We also found some unnecessary statements in the way that we were attempting to define BPCR, which we have now removed.

Comment 4: Also mention about how weighting been done. If possible provide stata syntax as a supplementary file.

Response: Thank you for your inquiry. The data were weighted by applying a weighting factor (v005/1000000) to minimize under- or over-representation of the data in the surveys due to differential selection among strata. Using the svyset command, the data was further structured as survey data. For more clarity for the readers, we have incorporated the statement in the ‘Data management and statistical analysis’ section of the "Revised Manuscript with Track Changes" on Page 7, Lines 181-185.

Comment 5: Formatting of tables and figures should be done as per the PLOS one requirement. Please review one published paper so that you can work accordingly. Please work on it. Also, take care of table and figure headings and presentation.

Response: we appreciate your insightful suggestion. We tried to present the captions for tables and figures as per the journal’s requirements.

Comment 6: Is this really the first study published from Ethiopia? You have mentioned about the BPCR message prevalence, I suggest these should be mentioned on your background section too. First paragraph is quite long, I suggest you to focus on to discuss about summary of your findings in this paragraph.

Response: thank you for your comment and suggestion. It might not be the first study in Ethiopia, and thus we removed the first statement from the discussion section and we merely focused on the findings. In addition, we tried to make the first paragraph more concise and precise.

Comment 7: Make your conclusion catchier. Don’t copy information from result section. Moreover, some of the information fits into the recommendation section in the discussion section.

Response: we entirely accept your suggestion and we have corrected it accordingly. the revised version is highlighted in the ‘conclusion’ section of the "Revised Manuscript with Track Changes" on Page 20, Lines 511-516.

I suggest you review the comments attached below for further improvement of the paper. Please review and revise the paper critically.

Response: Thank you for your constructive comments and suggestions, which we got of them as valuable input in the improvement of our manuscript. We received all of them as a valuable contribution to our ongoing work. In the following section, we tried to respond on all the possible comments and suggestions from both reviewers #1 and 2

END________________________________________

 THANK YOU!!!

Authors’ Response to Reviewer#1(Dear Rita Adhikari)

General comments: 

1. I enjoyed reading your paper however there are some room for the improvement. I have commented on the PDF version of your paper and highlighted the main part. However some others things need to be improved aside of the highlighted parts as follow: 

Response: Thank you very much for taking the time to review our work and for your positive feedback. We received your thoughtful, and generous review, along with helpful feedback and suggestions, as a valuable contribution to our ongoing work. We have tried to address all the possible comments and suggestions raised by you in the following session.

2. Please review the Plos One guideline intensively.

Response: thank you for your suggestion. During the initial submission, we entirely followed the ‘PLOS ONE's style requirements’ that was accessed at https://journals.plos.org/plosone/s/file?id=wjVg/PLOSOne_formatting_sample_main_body.pdf

3. Polish the language and Revise the formatting

Response: thank you for your suggestion. All the authors independently reviewed the overall format of the manuscript and we tried to amend the statements that need correction. We have highlighted them throughout the revised manuscript. Regarding the the formatting we have prepared the manuscript as per ‘PLOS ONE's style requirements’ that was accessed at https://journals.plos.org/plosone/s/file?id=wjVg/PLOSOne_formatting_sample_main_body.pdf

4. Reorganized some of the paragraph that I have highlighted in the Pdf file.

Response: We gave due emphasis for all the comments raised in the highlighted PDF and the responses were mentioned as follows.

Response to comments on Track change

Comment 1: regarding the data availability statement: Can you please mention what sorts of restrictions?.

Response: Thank you for asking. As the current study was based on DHS data, the ICF office made restrictions on sharing the data to other parties other than the authors mentioned in the study. All the statement is mentioned and highlighted in the ‘data availability’ section of the "Revised Manuscript with Track Changes", Line 527-530, Page 20. 

Comment 2: In the introduction section, Please confirm the full form of LMICs)

Response: thank you for your meticulous review. It was to mean Low and Middle Income Countries and We have corrected and highlighted it in the ‘Introduction’ section of the "Revised Manuscript with Track Changes", Line 66, Page 3. 

Comment 3: please do segregate this part which is written as ‘Study setting, data source, and study period’

Response: thank you for your insightful suggestion which enhances the clarity of the manuscript. Accordingly, we have tried to make more clear by changing the heading to ‘study design and setting’. The corrected version was highlighted in the ‘Methods and materials’ section of the "Revised Manuscript with Track Changes", Line 117, Page 4. 

Comment 4: Please do write this section again as the study population are not clearly mention , please give more background of the study population . Also, What do you mean by preceding over here, can you please make it more clear?What is source popualtion and their purpose? Please clearify the inclusion and exclusion criteria in a clear way.

Response: Thank you for your valuable suggestions and inquiries. We apologize for the ambiguous statement, and we have now attempted to figure out both the source and the study population. Regarding the source population, they are a large segment of the women population aged 15-49 years in which we finally generalize (infer) our findings. On the other hand, study populations belong to those who were directly dissected from the source population and fulfill eligibility criteria. We hope you will accept and get our explanation. The corrected version of the statement was highlighted in the ‘population’ section of the "Revised Manuscript with Track Changes", Line 126-128, Page 4.

Comment 5: How the validity and reliability of the questionnaire was confirmed?

Response: Thank you for asking. The EDHS employed standardized and validated data collection tools and procedures, which were explicitly explained in the 2016 EDHS report, and hence repeating the procedure isn't suggested.

Comment 6: in the ‘Measurement of variables of the study, reorganize the statement and Please, Confirm the question again

Response: Thank you for your in-depth review. We have tried to proofread the entire statement and corrected it accordingly. In addition, we made the outcome assessment questions more clear and highlighted them in the ‘Measurement of variables of the study’ section of the "Revised Manuscript with Track Changes", Page 5.

Comment 7: In the ethical consideration section, Would you please elaborate it more like how the consent from the one who is under 18 were taken if they are married and living separately from their parents? How you ensure their ethical measures? Please also add the citation.

Response: Thank you for asking. The assent was taken for individuals under the age of 18 from their families or guardians, and consent was granted for those over the age of 18. We have made the statement more clear and highlighted it in the ‘Ethical consideration and consent to participate’ section of the "Revised Manuscript with Track Changes", Line 259-260, Page 9.

Comment 8: In which table this information was mentioned??

Response: We have reported the overall proportion of missing all BPCR messages in the last column of the table as 44.4% and we kindly request you to crosscheck.

Comment 9: Regarding Figure 1, Please follow the Plos One guideline for the limitation of the figures

Response: Thank you for your suggestion. We have strictly followed the Plos One Manuscript submission guideline and we have checked the caption and formats of all figures accordingly.

Comment 10: in the discussion section, Which is your explanatory variable ??? 

access to radio or listen to radio?

Response: it was to mean women who never listen to the radio. We have corrected and highlighted it in the ‘Discussion’ section of the "Revised Manuscript with Track Changes", Line 428, Page 17.

Comment 11: Please take care of this formatting and rewrite the section in the last paragraph of the discussion section. 

Response: we appreciate your in-depth review. We have proofread the whole paragraph and made some amendments accordingly. The revised statements were highlighted in the ‘Discussion’ section of the "Revised Manuscript with Track Changes", Line 494-504, Page 19.

Thank you for your constructive comments and suggestions, which we got as valuable input in the improvement of our manuscript. We received all of them as a valuable contribution to our ongoing work. 

END_______________________________________

 THANK YOU!!!

Author's Response to Reviewer#2

General Comments

This is an interesting paper and the research question is of great practical significance. The following are some suggestions for revision.

Response: Thank you very much for taking the time to review our work and for your positive feedback. We received your thoughtful, and generous review, along with helpful feedback and suggestions, as a valuable contribution to our ongoing work. We have tried to address all the possible comments and suggestions raised by you in the following session.

Comment 1: As far as I know, OLS (Ordinary Least Squares) is most suitable for situations where the dependent variable is continuous. In this paper, it seems that the OLS model was used for modeling the relationship between "missing BPCR messages" and explanatory variables, whereas the "missing BPCR messages" is a binary variable. Please provide the rationality of adopting this method.

Response: thank you for your meticulous review and inquiry. Ordinary Least Squares (OLS) is a common statistical technique used in both spatial analysis and linear regression, but its application and interpretation can differ between these two contexts. Here's an overview of the key differences between OLS in spatial analysis and linear regression:

 In linear regression, the primary objective of OLS is to model the relationship between a continuous dependent variable and one or more independent variables whereas in spatial analysis, it is used to account for spatial dependencies or spatial autocorrelation in the data. It is used when the observations are not independent and may exhibit spatial patterns or spatial relationships that need to be addressed.

 OLS in Linear Regression, can be applied to any data where you want to model a linear relationship between variables, whereas, in spatial analysis, it typically deals with geospatial data, where the locations of observations or entities are important. 

 OLS in Linear regression, assumes that the observations are independent of each other whereas in spatial analysis it explicitly accounts for spatial dependencies (i.e. it incorporates the idea that nearby locations may have similar values for the dependent variable, and it attempts to model and control for this spatial autocorrelation).

 OLS in Linear Regression tells us the coefficients of the independent variables represent the change in the dependent variable for a one-unit change in the corresponding independent variable, holding all other variables constant. Whereas in Spatial Analysis with OLS, the interpretation of coefficients can be more complex. The coefficients often represent the change in the dependent variable associated with a one-unit change in the independent variable, while accounting for the spatial dependencies. Spatial lag or spatial error terms are often included in the model to account for spatial autocorrelation.

To summarise despite it has a similar name, their function is entirely different. The OLS that we used in spatial regression is a global test to select significant spatial predictors for the local( i.e. Geographical weighted regression (GWR)). It is not commendable to run the local(GWR) model without running the global(OLS) model in spatial regression. We hope you will accept our justification based on statsitcal grounds. 

Comment 2: There are some spelling errors in the paper. For example, "Ordinary List Squares" should be replaced by "Ordinary Least Squares". 

Response: We appreciate your meticulous review. The misnaming in the OLS was corrected and highlighted in the ‘Spatial regression analyses’ section of the "Revised manuscript with track changes", Line 232, Page 8

Comment 3: In table 2, the test statistic "X2" should be replaced by Greek letter χ, and the "2" would be marked as superscript.

Response: We accept your suggestion and we have corrected and highlighted it in Table 2 of the "Revised manuscript with track changes", Pages 10-11

Comment 4: In table 2, there are some calculation errors. For example, "126(43)" should be replaced by "126(43.3)"; In Somali, the total frequency is 11, whereas the sum of "Receipt of BPCR message" and "Not receipt of BPCR message" was inconsistent with the total. So please double-check the accuracy of each figure in the table.

Response: we greatly appreciate your meticulous review. We have checked the entire results and this was the only erroneous figure in the table. We have corrected and highlighted it in Table 2 of the "Revised manuscript with track changes", Page 10. 

Thank you for your constructive comments and suggestions, which we got them as valuable input in the improvement of our manuscript. We received all of them as a valuable contribution to our ongoing work. 

THE END____________________________________

 THANK YOU!!!

---

## [Editor Report · Decision Letter 1]

29 Nov 2023

Spatial variation and predictors of missing birth preparedness and complication readiness (BPCR) messages in Ethiopia: Spatial analyses

PONE-D-23-28170R1

Dear Dr. Habte,

We’re pleased to inform you that your manuscript has been judged scientifically suitable for publication and will be formally accepted for publication once it meets all outstanding technical requirements.

Kind regards,

Kanchan Thapa, MPH, MPhil

Academic Editor

PLOS ONE

Additional Editor Comments (optional):

Dear Authors,

Thank you for submitting the revised version of the paper. Prior to the formal publication of the paper, I suggest you to correct the following things.

1. Please remove Spatial analyses from the title. Already from the title it is clear that paper deals with the spatial analysis.

2. Line 98, it is better to write limited evidence rather than writing no exiting evidence.

3. Line 119-123, please provide the reference for your information.

4. Line 157, please remove unnecessary dots.
---

## [Editor Report · Acceptance letter]

1 Dec 2023

PONE-D-23-28170R1 

Spatial variation and predictors of missing birth preparedness and complication readiness (BPCR) messages in Ethiopia 

Dear Dr. Habte:

I'm pleased to inform you that your manuscript has been deemed suitable for publication in PLOS ONE. Congratulations! Your manuscript is now with our production department. 

Kind regards, 

on behalf of

Mr. Kanchan Thapa 

Academic Editor

PLOS ONE